# Oil Media on Paper: Investigating the Interaction of Cold-Pressed Linseed Oil with Paper Supports with FTIR Analysis

**DOI:** 10.3390/polym15112567

**Published:** 2023-06-02

**Authors:** Penelope Banou, Stamatis Boyatzis, Konstantinos Choulis, Charis Theodorakopoulos, Athena Alexopoulou

**Affiliations:** 1Department of Conservation of Antiquities and Works of Art, University of West Attica, Egaleo, 12243 Athens, Greece; 2Department of Arts, Northumbria University, Newcastle, Newcastle upon Tyne NE1 8ST, UK

**Keywords:** cold-pressed linseed oil, alkaline-refined linseed oil, stand oil, pure cellulosic paper, lignocellulosic paper, ATR-FTIR, reflectance FTIR

## Abstract

Previous works of the authors have presented the changes in the optical, mechanical, and chemical properties of the oiled areas of the supports that occur upon ageing due to oil-binder absorption in works of art on paper and printed material. In this framework, transmittance FTIR analysis has indicated that the presence of linseed oil induces the conditions to promote the deterioration of the oil-impregnated areas of the paper supports. However, the analysis of oil-impregnated mock-ups did not provide detailed information about the input of linseed oil formulations and the different types of paper support on the chemical changes that occur upon ageing. This work presents the results of ATR-FTIR and reflectance FTIR, which were used for compensating the previous results, proving indications on the effect of different materials (linseed oil formulations, and cellulosic and lignocellulosic papers) on the development of chemical changes, thus, on the condition of the oiled areas upon ageing. Although linseed oil formulations have a determining effect on the condition of the oiled areas of the support, the paper pulp content appears to have an input to the chemical changes that occur in the system of paper–linseed oil upon ageing. The results presented are more focused on the oil-impregnated mock-ups with cold-pressed linseed oil since results have indicated that this causes more extended changes upon ageing.

## 1. Introduction

Books, archival material, and works of art on paper present specific types of damage, when the oil binders in oil paints and traditional oil-based inks are absorbed by the paper supports, such as discolouration, loss of mechanical strength, and drop of pH. These problems have been recorded in diverse case studies, differing in intensity and extent, setting questions for the input of materials, paper, and oil type [1,2,3,4,5]. The combination of paper and oil could raise substantial concerns for the preservation of works upon ageing since the condition of the oiled areas of the support could deteriorate, and these could break into pieces, locally, but also overall [6].

As far as works of cultural heritage are concerned, only limited research on this matter has been reported until recently [7,8,9,10]. Although there is research on the effect of mineral oils used for the insulation of Kraft paper in transformers [11,12,13,14,15], the results could not be applied to printed material and works of art, where drying oil binders are used as oil binders. In addition, Kraft paper could not reflect the variety of paper supports used for works of art, archival material, and books.

Recently, the authors of this work have published results of the study on the changes in the optical, mechanical, and chemical properties that occur on the system of paper–linseed oil upon ageing, aiming to the interpretation of damage recorded on works of art and printed materials due to oil-binder absorption [16,17]. The outcome of colourimetry, opacity, pH, and tensile strength measurements indicated that linseed oil has a predominant role in these changes, highlighting that the different linseed oil formulations and paper types have an input to the differences noted between the different sets of mock-ups. Transmittance Fourier-transform infrared spectroscopy (FTIR) analysis supported the results derived by the other methods of research and also indicated that the presence of linseed oil can induce the conditions to promote the deterioration of the oil-impregnated paper mock-ups upon ageing, as it has been suggested by the intense drop of pH of the oil-impregnated mock-ups from the initial stages of ageing and the reduction of the mechanical strength upon ageing. It also indicated that there were variations in the extent of oxidation and hydrolysis in the different series of mock-ups when the combinations of linseed oil formulations and paper types differ [16,17].

However, that work did not provide facts on the input of the three linseed oil formulations, or that of the paper content, on the chemical changes of the system of paper–linseed oil upon ageing, which is the aim of the present work. Therefore, the objective of the applied experimental methodology is to provide a hypothesis on the causes of the differences recorded between the various sets of mock-ups in their chemical behaviours, and, furthermore, their optical and mechanical changes of properties upon ageing. So, the results could contribute to the interpretation of the diversity in the condition recorded on the oiled areas of the supports of works of art, but also to the condition assessment of the category of oil media on paper.

The present work will focus on the results delivered by attenuated total reflectance (ATR) FTIR and reflectance FTIR analysis to complement those of transmittance FTIR. ATR-FTIR was conducted to investigate the chemical changes of the three linseed oil formulations individually, namely, that of cold-pressed linseed oil, alkaline-refined linseed oil, and stand oil, that submitted to the same accelerated ageing conditions as the oil-impregnated mock-ups (80 °C and 77% RH). In addition, reflectance FTIR was used to analyse the dried linseed oil films that were extracted from the oil-impregnated mock-ups. Since cold-pressed linseed oil presented similar changes to those of refined linseed oil, which were more extended than stand oil, the results on oil-impregnated paper mock-ups with cold-pressed linseed oil, made of three different types of paper, are discussed in comparison with those of neat linseed oil mock-ups and the oil films resulted from oil extraction (details of the products used are available in the Appendix A Appendix A).

A pure cellulosic and two lignocellulosic papers were used for the mock-ups to investigate whether the absence, removal, or presence of lignin content in the pulp has an effect on the system of paper–linseed oil. Although the presence of lignin has been regarded as responsible for the deterioration of the supports in the past, recent studies have indicated that although it causes the paper to be light-sensitive when it is present, it may act as a hindered phenolic antioxidant [18]. In addition, the function of the network of lignin molecules provides a mechanically strong composite material with cellulose fibres [19].

Finally, qualitative and quantitate processing of the results were performed to investigate the differences recorded upon ageing between the sets of oil-impregnated paper mock-ups.

## 2. Experimental

### 2.1. Materials

Three formulations of linseed oil and three types of paper support were used to prepare several series of mock-ups, employing standardised models prepared to serve the purposes of this study. Linseed oil was selected as the most representative oil binder used in printmaking and oil painting until the 19th century. The three types of linseed oil were selected that differ in the method of manufacture, namely, cold-pressed linseed oil (CP), alkaline-refined linseed oil (RF), and stand oil (StL). Cold-pressed linseed oil was selected as the purest type of linseed oil since it has not been subjected to a thermal or chemical process; alkaline-refined linseed oil is the most commonly used type for oil painting, which is thermally and chemically processed during manufacture, and stand oil, which is pre-polymerised through long-term processing in high temperature in the presence or absence of air. The difference in the methodology of oil manufacture provides these formu lations with different physicochemical properties, such as wetting power, drying rate, yellowing or darkening, viscosity, rheology, acid value, and degree of polymerization [20,21,22,23,24,25].

Mock-ups were made of pure cellulosic paper and two types of lignocellulosic papers (with different lignin content). In particular, the paper described as “Cotton” paper (C), is made of pure cotton linters, and it is unbuffered, with no fillers or sizing, 80 g/m^2^. This paper type provides the purest form of paper. For the lignocellulosic papers, a typical watercolour paper, “Montval” (M) white colour, cold-pressed, acid-free, made of wood pulp (soft and hardwood fibres) and limited lignin content, without optical brighteners or additives, 185 g/m^2^, and a wrapping paper, described as “Kraft” (K), brown colour, 100 g/m^2^, buffered, made of soft and hardwood fibres, containing lignin, fillers, additives, and metallic contamination.

The pulp content (fibres, lignin, sizing, and additives) and the physical properties (colour, weight, and surface characteristics) of these papers responded to the characteristics of the supports recorded in original case studies of works presenting problematic oiled areas, and represent a vast majority of the papers used for the creation of works of art and printed material through time [9]. The use of lignocellulosic supports refers to the paper supports mostly used from the mid-19th century until today.

### 2.2. Preparation of Mock-Ups and Artificial Ageing

Paper strips, 2.5 × 10 cm (width × length), comprising the three paper types were cut for the preparation sets of mock-ups, for the series of plain papers and those of oil-impregnated ones. For the sets of oil-impregnated mock-ups, paper strips were impregnated with 0.3 mL of the three types of linseed oil, respectively, using a 1 mL syringe. The volume of oil was adequate for uniformly impregnating the paper strips of all types, without leaving excess.

After 40 days of air drying at room conditions of 22 °C and 52% RH, in the dark (stage of ageing 0/40, while the stage of ageing 0 indicates that the mock-ups have not been subjected to drying or any other process), the mock-ups were subjected to artificial ageing in the dark, with controlled conditions of 77% relative humidity (engaging saturated aqueous solution of sodium chloride, 0.15 g/mL) and temperature of 80 °C in airtight vessels for 2, 4, 7, 14, 21, and 28 days (Figure 1). This methodology is recommended by the Library of Congress preservation department for ageing papers involved in experimental procedures, and it has been used in previous relevant works [9,16,17,18,26]. To investigate the chemical changes of the three linseed oil formulations, 0.3 mL of each linseed oil formulation was applied on a glass vials slide, one for each ageing stage. These were submitted to the same ageing conditions as for the plain paper and oil–impregnated mock-ups. In total, 15 sets of mock-ups were prepared to serve the purposes of research, as presented in Table 1.

For the study of the possible interaction of the linseed oil and paper support upon ageing, after the end of artificial ageing, oil was extracted by the oil-impregnated paper strips using chloroform, and then both the solution and the paper were submitted to FTIR analysis. For oil extraction, a piece of the oil-impregnated mock-ups (size of 1 × 2 cm) was immersed in 1 mL of chloroform in a glass vial and then placed in an ultrasonic bath sonicator for 20 min.

### 2.3. Methodology

Analysis of oil mock-ups was performed on a Bruker Alpha II FT-IR equipped with a Bruker diamond crystal ATR spectrometer (manufacturer’s spectral range 7500–400 cm^−1^), in absorbance mode, in the 4000–400 cm^−1^ region, at a resolution of 4 cm^−1^, and by summing 24 scans. The spectra were recorded on Bruker OPUS v.8.5 software and they were further processed on Spectragryph v.1.2.16. Normalisation correction was applied in all spectra. Second-order derivative FTIR spectra were studied to resolve overlapping peaks in which molecular groups with carbonyl bands contributed within a wide wavenumber range from 1760 to 1695 cm^−1^ and, in particular, the peaks at c.1745 cm^−1^ due to carbonyls in glycerol ester linkages, and at c.1709 cm^−1^ due to carbonyls in carboxylic acid groups [26,27,28,29,30]. Second-derivative spectroscopy is a technique which enhances the separation of overlapping peaks of two or more components, a discrimination in favour of the sharpest features of a spectrum, and is used to eliminate interferences by broadband constituents [31,32].

For the ATR-FTIR analysis, the plain paper mock-ups (without the oil application) and the oil-impregnated paper mock-ups were placed directly on the diamond crystal (see the Appendix A, Appendix A). The neat linseed oil formulations were scraped off the glass surface with a scalpel 10A and placed on the ATR diamond crystal. For the fresh linseed oil formulations, a small droplet was applied to the diamond cell.

The performance of the analysis had to deal with the physical changes on the mock-ups upon ageing. At the first stages of ageing, an oil film covers the whole surface of the oil-impregnated mock-ups, while, in the core of the paper, the fibres and oil form a united mass. Gradually, the oil film becomes more “solid”, but it slowly recesses into the fibre net. After 14 days, the oil turns into a gel and movable form. Similar changes were recorded in the mock-ups of plain linseed oil formulations [16,17]. All these alterations imposed different variations in the application of the methodology (i.e., application of pressure).

Bruker Alpha II FT-IR was also used for reflectance FTIR. This was used only for the cold-pressed linseed oil extracted from oil-impregnated mock-ups. A droplet of the solution derived from the oil-extraction procedure was applied to a golden mirror substrate and allowed the solvent to evaporate. Analysis was conducted on the fine cold-pressed oil film formed on the golden mirror (transflectance) (see the Appendix A, Appendix A).

For all forms of analysis, and all series of mock-ups, measurements were taken from three different mock-ups at each ageing stage.

## 3. Results Discussion 

Research work on this matter resulted in a plethora of results. Only the findings that strongly support the objectives of this work are presented. Supportive information on the results has been included in the Appendix A. Please note that the tables and figures included in the Appendix A bear the characteristic symbol S. 

### 3.1. Plain Papers

ATR-FTIR analysis showed the profile of a pure cellulosic paper (cotton-based) for the “Cotton” paper and a typical lignocellulosic paper profile for “Montval” and “Kraft” papers, indicating the removal of lignin for “Montval”. The results were analogous to those delivered by transmittance FTIR analysis [16,17].

In particular, ATR- FTIR analysis of Cotton paper showed absorption at the band of 999 cm^−1^, as well as the C-C ring breathing band at ~1105 cm^−1^ and the C-O-C glycosidic ether at ~1155 cm^−1^, which are assigned as fingerprints of cellulose in paper [27,33,34,35,36,37,38,39]. Other characteristic bands that presented absorption and are related to the chemical structure of cellulose were hydrogen-bonded OH stretching at around 3600–3000 cm^−1^, CH stretching at 2917 cm^−1^, and CH wagging at 1316 cm^−1^ (Figure 2). The FTIR spectrum delivered was representative of cellulosic fibres, based on previous works (see Appendix A).

ATR-FTIR analysis of Montval and Kraft paper showed a typical lignocellulosic paper profile (Figure 1). For the Kraft paper, the presence of wood pulp was indicated through the lignin absorption at about 1590, 1505, 1450, ~1265, ~900, and 808 cm^−1^, based on the results of previous works (Appendix A). For the Montval paper, the removal of lignin was indicated by the lack of all lignin markers, while it showed characteristic absorption at 1202–1204, 1050, and 1030 cm^−1^ [27]. ATR-FTIR analysis, additionally, confirmed the presence of CaCO_3_ in both papers by the marked absorptions at about 1430 and 874 cm^−1^, along with clay, shown at 1030–1000 and 910 cm^−1^, typical of aluminum silicates [27] (Figure 2).

ATR-FTIR spectra of plain Cotton, Montval, and Kraft mock-ups did not display notable changes upon ageing (0–28 days of ageing). This was evident in the presentation of ATR-FTIR spectra on an overlay form (see Appendix A in Appendix A).

### 3.2. Neat Linseed Oil Formulations

The ATR-FTIR spectra of the fresh uncured cold-pressed linseed oil (CP) and refined linseed oil (RF) formulations (stage 0 days of ageing) were similar, showing the typical pattern of unsaturated triglycerols, which is typical for drying oils, and linseed oil respectively (see Appendix A) [20,27]. The spectra of the CP and the RF appear to overlap completely (Figure 3). In this stage, no difference was recorded between the two formulations of linseed oil.

In particular, they present absorption bands in the characteristic carbonyl band at 1746 cm^−1^ and the C-O stretching pattern at 1239, 1164, and 1101 cm^−1^, which are diagnostic for triglyceride ester linkages. Also, the band of olefinic C-H stretching band at 3010 cm^−1^ is attributed exclusively to unconjugated, symmetrically disubstituted cis double bonds as expected for the fatty ester composition. There are also bands of the cis CH out-of-plane deformation at 722 cm^−1^, the cis -HC-CH- stretching vibration at 1653 cm^−1^, the CH deformations at 1000–800 cm^−1^, and the C-C stretching vibrations at 1680–1600 cm^−1^ [27].

The ATR-FTIR spectrum of the fresh and uncured stand oil (StL) was quite similar to those of CP and RF, since several bands overlap, as shown in Figure 3. However, the spectrum presented a comparatively lower absorption in the bands 3010 and 1653 cm^−1^, which is commonly observed in the first stages of autoxidation (after an induction period) with the transformation of *cis*-to-*trans* double bonds, described as an isomerisation process [27]. This is consistent with the presence of an absorption band in 977 cm^−1^, which reflects the formation of *trans* C=C bonds. In addition, lower absorption with limited variations was observed in absorption between 1460 and 400 cm^−1^ (Figure 3), with a rather distinct one in the band of 723 cm^−1^, that responds to *cis* (C-H) out-of-plane deformation [27,28]. The profile of the ATR-FTIR spectra responds more to that of saturated triglycerols (see Appendix A), while the differences between StL, CP, and RF could be attributed to the extended heating in high temperatures of stand oil during manufacture, as suggested by other researchers [20].

Drying and hardening of linseed oil occur through the polymerization of triglycerides via autoxidation, a process involving free radicals. Free-radical polymerization reactions develop in three main stages, namely, the initiation, propagation, and termination, resulting in changes in the chemical structure of linseed oil [27,28,40]. However, polymerization competes with degradation processes, such as scission reactions, decomposition of unstable crosslinks, and hydrolysis [40].

After 40 days of drying in room conditions in the dark (stage 0/40 of ageing), ATR-FTIR spectra of the linseed oil formulations showed changes in absorption bands, as shown in Figure 3. It should be mentioned that the oil films of the three linseed oil formulations were already touch dry. The spectra of the three linseed oil formulations presented a comparable pattern of absorption, with variations in intensity. There were lower absorptions for CP and RF in the band of 3010 cm^−1^, broadening of the band 1743 towards 1700 cm^−1^, and an increase in absorptions of between 1350 and 700 cm^−1^. These reflect the initial chemical reactions involved in the drying of fresh linseed oil, mainly due to autoxidation, involving cross-linking reactions taking place between the triacylglycerols [18,26,27]. Due to these changes, the absorption spectra of the three linseed oil formulations became similar, with those of RF and StL overlapping in the region 3100–1375 cm^−1^ (Figure 3). Comparatively, cold-pressed oil appeared to have more intense changes between 0 and 40 days of drying, while variations for stand oil were more limited, as shown in Figure 3 and Figure 4. It could be suggested that the absence of heat and chemical pre-treatment of cold-pressed linseed oil, induced a more intense chemical condition. On the other hand, pre-polymerisation processing of stand oil has as a result lower absorption of oxygen, thus, slower development of oxidation reactions [21]. This observation was supported by the calculation of the integral ratio of the derivatives bands 1820–1570 cm^−1^ and 1450–400 cm^−1^ of the fresh linseed oil formulations dried for 40 days by those freshly applied, which presented the most characteristic changes upon ageing, and they are associated with the oxidative degradation and polymerization of oils (see Table 2 and Appendix A).

Upon ageing of the three linseed oil formulations, CP, again, presented the most intense changes (Figure 5). These developed with a gradual broadening of the band up to the 7th day of ageing, while on the 14th day of ageing the peak of 1738 cm^−1^ and that of 1710 cm^−1^ were equal, and, at the final stages, the peak of 1710 cm^−1^ was increased to a higher level than the peak at 1738 cm^−1^. Broadening of the specific band indicates the formation of carbonyl-containing species, such as aldehydes, ketones, and carboxylic acids, that occur during oxidative polymerisation, while the increase in absorption at the peak of 1710 cm^−1^ corresponds to the formation of carboxylic groups, thus acids, indicating the hydrolysis of the oil after the 14th day of ageing [20,27,28,40,41,42]. The development of these peaks was clearly observed after 28 days of ageing in the derivatives of the band 1760–1710 cm^−1^ (Figure 6 and Appendix A).

RF mock-ups presented equivalent changes (Figure 7). It could be only suggested that they were slightly more intense in the final stages due to the broadening of the band 1760–1696 cm^−1^. Finally, for StL, these changes developed at a more limited extent, so those recorded in the spectrum of the 28th day of ageing corresponded to those of the 14th day of ageing for the other two oil formulations (Figure 8).

The development of oxidative degradation, and the possible differences between the linseed oil formulations, were studied by calculating the ratio of the integration of the band 1900–1550 cm^−1^ by that of the hydrocarbons band 3200–2800 cm^−1^ (see Appendix A and Figure 9), following similar practices used by other authors to indicate the oxidation status of the oil films [31]. These results indicated that more intense changes occur at the first stages of ageing, and CP and RF present a similar trend, while the changes for StL were more subtle throughout ageing (Figure 9). 

Additionally, the difference in the increase in absorption at 1710 cm^−1^ was studied by calculating the ratio of integration of the band 1730–1695 cm^−1^ by that of 1760–1730 cm^−1^ (see Appendix A). The graphic representation of the results confirmed the aforementioned observations (Figure 10).

Since CP and RF presented similar changes upon ageing, and much more extended in comparison with StL, the interpretation of the results was focused on the mock-ups impregnated with cold-pressed linseed oil, which are presented in the following sections.

### 3.3. Cold-Pressed Oil Extractions

The reflection-FTIR spectra of CP extracted from the Cotton oil-impregnated mock-ups (CP/C) presented the most significant changes on the band 1700–1650 cm^−1^ at all stages of ageing. The spectrum of the extracted CP from the oil-impregnated mock-up after 40 days of drying (stage 0/40) indicated that the autoxidation had already been initiated, taking into consideration the broadening of the band between 1750–1700 cm^−1^. There were limited changes up to the 21st and 28th day of ageing, where the peak at 1710 cm^−1^ appeared to increase higher than that of 1741 cm^−1^, indicating the hydrolytic and oxidative degradation of the CP (see Figure 11a and Appendix A). From 0 to 14 days of ageing a peak at 1720 cm^−1^ was also recorded, possibly indicating the formation of saturated ketones, which was not evident in the analysis of neat linseed oil formulations. Likewise, a constant but limited noise was recorded between 1700 and 1450 cm^−1^.

The reflectance spectra of the CP extracted from Montval oil-impregnated mock-ups (CP/M) at the several stages of ageing presented differences between that of CP extracted from Cotton oil-impregnated mock-ups. Again, the most significant changes were recorded between 1750–1710 cm^−1^. Although there is a broadening of the region, the band at the 1710 cm^−1^ increased, but it only reaches slightly higher than that of 1738 cm^−1^ (Figure 11b and Appendix A). There were also peaks between 1735 and 1710 cm^−1^, almost at the same level. Also, there was an intense “noise” between 1700–1462 cm^−1^. All these extra peaks indicate the development of chemical actions that did not occur in neat linseed oil formulations. It could be suggested that CP have reacted with the pulp content materials, such as the calcium ions (Ca^+2^_)_ (Ca(OH)_2_) is used in the paper-making process) providing a slightly diverse chemical profile. It could be suggested that the peaks responded to the presence of fatty acid metal soap. Fatty acid metal soap formation on the painting layers of oil paintings on canvas, and their effect on their preservation, is a known research issue [43,44,45,46,47,48,49,50]. Metal soap formation occurs during the neutralisation of the fatty acids produced by the hydrolytic degradation of oil binders, when the latter react with metal ions in metal hydroxides, oxides, and mixed hydroxide salts, creating an ionomeric-like structure [27,43,44]. Calcium fatty soaps have been identified, among others, in previous works [27,43,51]. Further work is needed to investigate the formation of possible by-products due to the reaction of linseed oil and paper pulp contents.

Similar observations were made for CP extracted from Kraft oil-impregnated mock-ups (CP/K). In this case, the peaks in 1719 and 1708 cm^−1^ reached the same level as that at 1739 cm^−1^ (Figure 11c and Appendix A). Detail images and the derivatives of the band of 1700–1650 cm^−1^ confirmed the formation of a separate peak at 1720 cm^−1^, but also of others, presenting a different degree of absorption intensity among the sets of impregnated mock-ups (Figure 12 and Appendix A).

The results provided initial indications that paper, and, in particular, pulp content, has an input to the development of chemical changes in the system paper–linseed oil. This hypothesis could provide explanations for the differences recorded in spectra of CP derived from the extraction of different paper-type impregnated mock-ups. It should not be ignored the lack of alkaline buffer in Cotton’s paper pulp and the presence of lignin, but also of impurities (such as traces of metals) in Kraft’s one. The presence of an alkaline buffer, and, possibly, that of lignin, in the paper pulp, could restrain the production of acids or could induce variant conditions for the evolution of other chemical reactions and by-products, consequently.

### 3.4. Oil-Impregnated Mock-Ups

The spectra of Cotton mock-ups impregnated with CP presented more characteristic changes in the bands that respond to carbonyl-containing species (such as aldehydes and ketones) and carboxyl acids (1600–1750 cm^−1^), which are associated both with cellulose oxidation as well as the oxidation of linseed oil [27,28,35,36,37,38,39,40,41,42]. Even at stage 0 (after 40 days of drying), the broadening of the band 1750–1600 cm^−1^, suggests that the system was in an oxidated stage (Figure 13). This implies that the application of CP even on the purest form of paper could promote oxidation reactions in the system of paper–linseed oil within 40 days. This can be attributed to the oxidative degradation of linseed oil that provides the right conditions for the oxidation of cellulose or the oxidation of the system paper–linseed oil. This band presented a gradual increase in absorption upon ageing. After seven days of ageing, the absorption at the peak of 1710 cm^−1^ increased higher than that of 1735 cm^−1^, implying conditions of deterioration of the system paper–oil (Figure 13). It could be suggested that the lack of sizing, alkaline buffer, or other additives allowed the cold-pressed linseed oil to be thoroughly introduced to the fibre net and, possibly, in duced chemical reactions in the system without having any factor to restrain them. The results have confirmed the dramatic decrease in the tensile strength of the Cotton oil-impregnated mock-ups with CP upon ageing [16].

Montval and Kraft oil-impregnated mock-ups presented similar behaviour with those of Cotton at the first stages of ageing, but the increase in absorption in the band 1750–1600 cm^−1^ appears to be less intense. Also, the increase in absorption at 1710 cm^−1^ is more limited, as it did not reach higher than that of 1738 cm^−1^ for the Montval mock-ups at the final stages of ageing, while the same one appeared to be slightly higher for the Kraft mock-ups (Figure 14 and Figure 15). These observations were supported by the calculation of the integral ratio of the band 1730–1695 cm^−1^ by that of 1760–1730 cm^−1^ (Figure 16, Appendix A). It could be suggested that the Montval and Kraft mock-ups did not reach the same stage of deterioration as those of Cotton. This hypothesis could also support the results of tensile strength measurements that showed lesser mechanical-strength reduction for Montval and Kraft oil-impregnated mock-ups [16].

However, Montval and Kraft oil-impregnated mock-ups showed the evolution of an absorption band at 1650–1540 cm^−1^, which was more evident at the final stages of ageing. This could be attributed to the formation of carboxylates or fatty acid metal soaps, such as calcium or aluminum soaps, formulated by the reaction of free fatty acids with the compounds of the alkaline buffer and additives in the paper pulp. This could explain the lower absorption of the peak at 1710 cm^−1^ at the final stages of ageing, in comparison to that of the neat cold-pressed linseed oil, since fatty acids were possibly consumed for the soap formulation.

For all sets of mock-ups studied in the present work, ATR-FTIR spectra did not provide clear indications for the changes that occur to other bands upon ageing. Transmittance FTIR has indicated reduction for the bands that respond to volatile oxidation compounds (2855–2853 cm^−1^ and 2810 cm^−1^), hydroperoxides, and alcohols (3200–3600 cm^−1^ and 1100–1210 cm^−1^), and the formation of conjugated bonds (such as in 1624, 1633, 950, and 723 cm^−1^) and oxidative polymerization (1099–1238 cm^−1^) in the final stages of ageing. All these could be attributed to the degradation of linseed oil according to the FTIR analysis of linseed oil subjected to different accelerating and storage conditions reported by other researchers [27,28,29,30].

The spectra of the oil-impregnated mock-ups were compared with those of the plain papers, the neat linseed oil formulations, and those derived from oil extraction. It was evident that CP had the dominant effect on the evolution of chemical changes. However, more similarities in the evolution of chemical changes were observed between the oil-impregnated mock-ups with CP and those of CP extracted from the three types of papers, respectively. It could be suggested that the evolution of the absorption band 1650–1450 cm^−1^ on the spectra of oil-impregnated mock-ups possibly responds to the noise recorded on the band 1650–1450 cm^−1^ of the spectra derived from extracted CP from Montval and Kraft oil-impregnated mock-ups, possibly due to the same provenance, the presence of alkaline buffer, and other additives. Finally, the derivatives on the band 1745–1740 cm^−1^ for all sets of oil-impregnated mock-ups also showed the formation of a peak at 1720 cm^−1^, recorded in oil-extraction spectra, presenting different absorption intensities. Two separate peaks were also recorded in the band of 1745–1730 cm^−1^ for Cotton and Montval oil-impregnated mock-ups in 1747 and 1738 cm^−1^ (Figure 17).

### 3.5. Paper after Oil Extraction

Oil extraction from oil-impregnated mock-ups required repeated extractions with chloroform to provide samples without residues of CP. Cotton oil-impregnated mock-ups presented the most extended chemical changes in comparison to those of Montval and Kraft, which follows with the aforementioned results, so these were selected to demonstrate the chemical changes of the paper support after oil extraction.

The Cotton paper mock-ups, after CP oil extraction, presented mostly changes in the carbonyl-containing band (1740–700 cm^−1^). From the 4th day of ageing and onwards, a gradual increase in absorption on the band between 1700–1750 cm^−1^ was recorded, indicating the development of the oxidation of paper, while on the 21st day of ageing, the absorption at 1710 cm^−1^ increased and appeared to give a higher peak than the rest of the band (Figure 18). These changes could be attributed to the oxidation and oxidative hydrolysis of cellulose according to other researchers’ results [52,53]. The hydrolysis of CP and the consequent formation of fatty acids created the conditions for the development of these chemical changes.

## 4. Conclusions

This work aimed to investigate the input of the different linseed oil formulations and paper types to the chemical changes that occur to the oil-impregnated mock-ups upon ageing and possible interactions between linseed oil and paper content that could influence their development.

Results showed that linseed oil is the dominating factor in the evolution of chemical reactions upon ageing. The results indicated that the most significant changes appear in the band 1750 to 1700 cm^−1^. The ATR-FTIR of neat cold-pressed and alkaline-refined linseed oil presented similar changes upon ageing, much more extended in comparison with those recorded for stand oil. The process of manufacturing alkaline-refined linseed oil has a limited effect on the oxidative polymerization and degradation of the oil upon ageing. Pre-polymerisation of stand oil has a significant effect on the chemical changes upon ageing. Hence, the chemical changes of the paper–linseed oil system could be associated with the extent of those occurring to the linseed oil formulations, respectively. This has been indicated by the results obtained from the measurement of optical and mechanical properties in previous work by the authors [16,17].

However, it was the results of the cold-pressed linseed oil extracted from the oil-impregnated paper mock-ups that indicated that the paper pulp content has an input to the development of chemical reactions. These results also supported the interpretation of those derived from the oil-impregnated cotton mock-ups that presented common changes in certain areas of the FTIR spectra. It has been suggested that the presence of alkaline buffer in the paper pulp, possibly, constrains the formation of acids, that are, possibly, consumed to formulate metal soaps of fatty acids. The outcome of this work provides additional data for the variations in the extent of deterioration of oil-impregnated papers for the first time.

The results could not provide a secure hypothesis of the input of lignin to the chemical changes. Both lignocellulosic papers bear an alkaline reserve, so they present fewer extended chemical changes than oil-impregnated pure cellulosic paper. However, the oxidation of oil-impregnated Montval paper, in which lignin has been removed, appears to be even less than that of Kraft, which included lignin in its pulp content. Nevertheless, the additives and impurities in Kraft’s pulp content could also have an input in that. 

Thus, the applied methodology managed to complement previous results on the factors that influence the condition of the oiled areas of the support of archival and printed material, books, and works of art on paper.

## Figures and Tables

**Figure 1 polymers-15-02567-f001:**
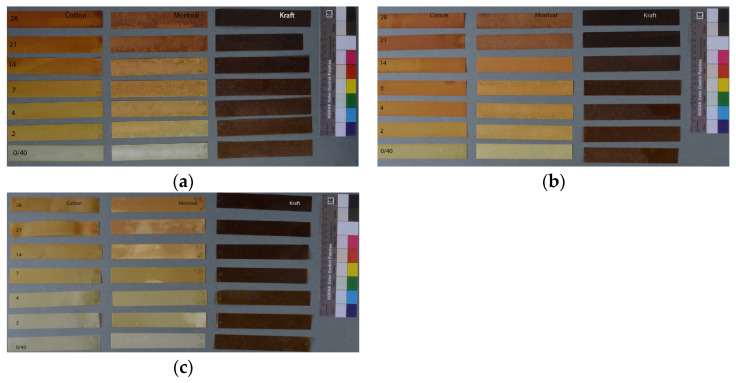
Sets of oil-impregnated mock-ups at the several stages of ageing (0/40, 2, 4, 7, 14, 21, and 28 days) of all paper types: (**a**) oil-impregnated mock-ups with cold-pressed linseed oil, (**b**) oil-impregnated mock-ups with refined linseed oil, and (**c**) oil-impregnated mock-ups with stand oil. Cotton mock-ups on the left, Montval mock-ups in the centre, and Kraft mock-ups on the right.

**Figure 2 polymers-15-02567-f002:**
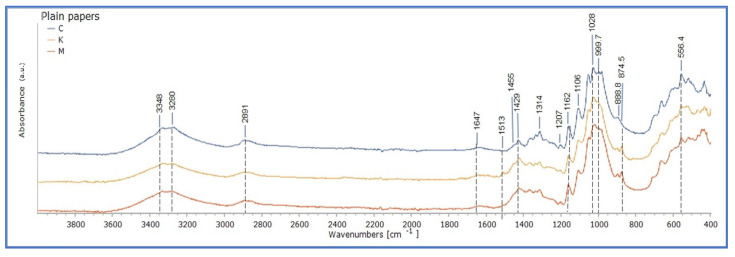
ATR-FTIR spectra of plain Cotton (C), Montval (M), and Kraft (K) mock-ups before ageing (stage 0).

**Figure 3 polymers-15-02567-f003:**
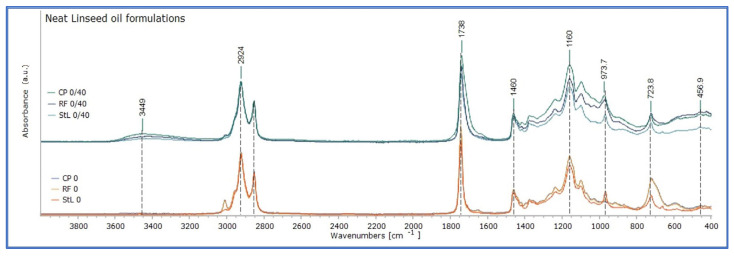
The FTIR spectra of the three linseed oil formulations (CP, RF, and StL), fresh and uncured on the lower part of the graph (0 days), and after 40 days of drying on the top part of the graphs (0–40 days).

**Figure 4 polymers-15-02567-f004:**
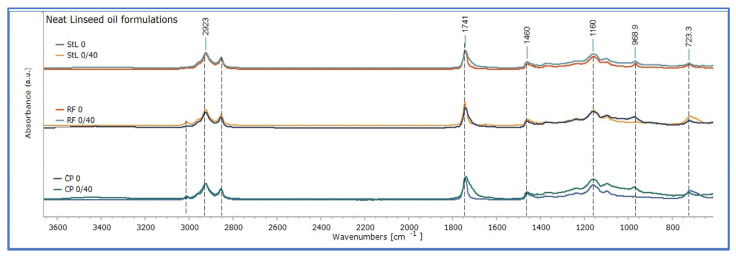
The FTIR spectra of the three linseed oil formulations, fresh and uncured, and after 40 days of drying, in sets for each formulation. On the top, cold-pressed linseed oil (CP), in the middle alkaline-refined linseed oil (RF), and at the bottom, stand oil (StL).

**Figure 5 polymers-15-02567-f005:**
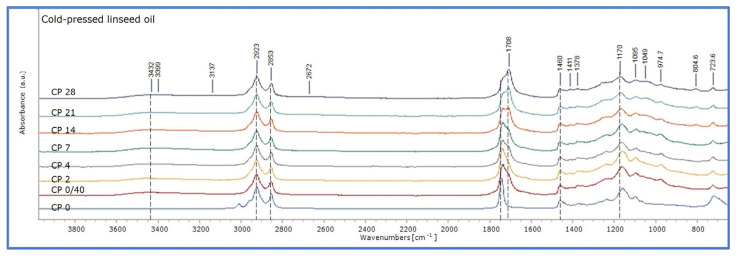
ATR-FTIR spectra of cold-pressed linseed oil mock-ups (CP) at all ageing stages (0–28 days of ageing).

**Figure 6 polymers-15-02567-f006:**
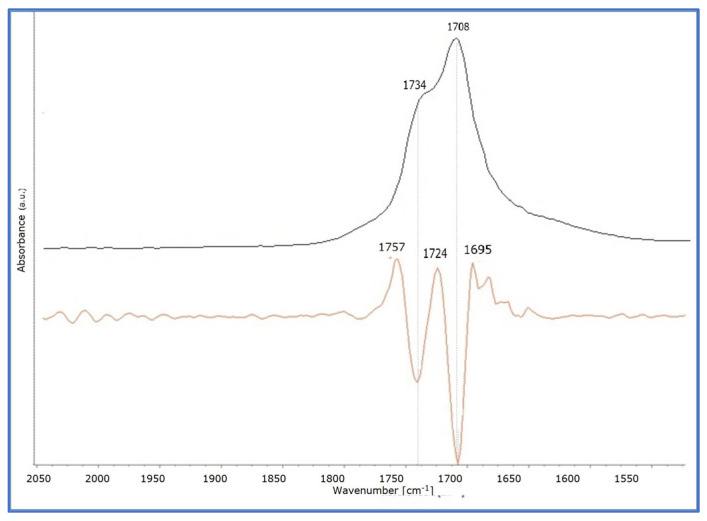
The ATR-FTIR spectra of CP at 28 days of ageing (on the **top**, blue line) and the derivatives (at the **bottom**, yellowish line). The development of two different peaks with the band of 1760–1700 cm^−1^ at 1738 and 1708 cm^−1^ is observed.

**Figure 7 polymers-15-02567-f007:**
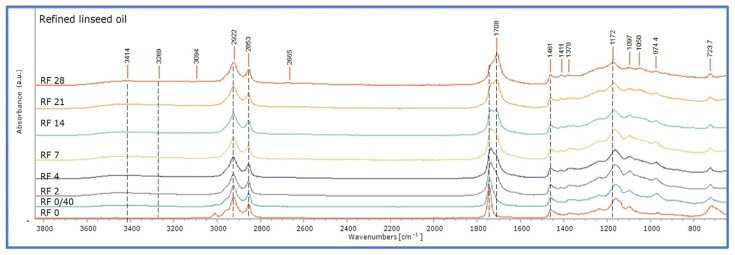
ATR-FTIR spectra of refined linseed oil mock-ups (RF) at all ageing stages (0–28 days of ageing).

**Figure 8 polymers-15-02567-f008:**
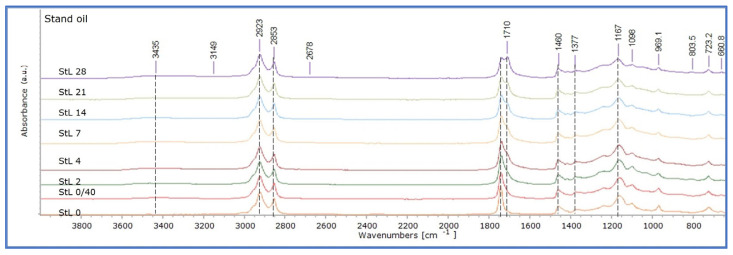
ATR-FTIR stand oil mock-ups (StL) at all ageing stages (0–28 days of ageing).

**Figure 9 polymers-15-02567-f009:**
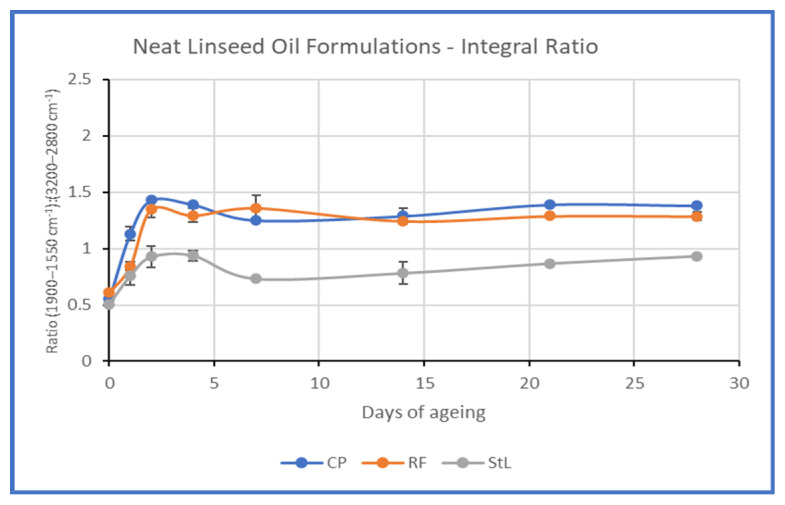
Graphic representation of the integral ratio of the band 1900–1550 cm^−1^ by that of the hydrocarbons’ band 3200–2800 cm^−1^.

**Figure 10 polymers-15-02567-f010:**
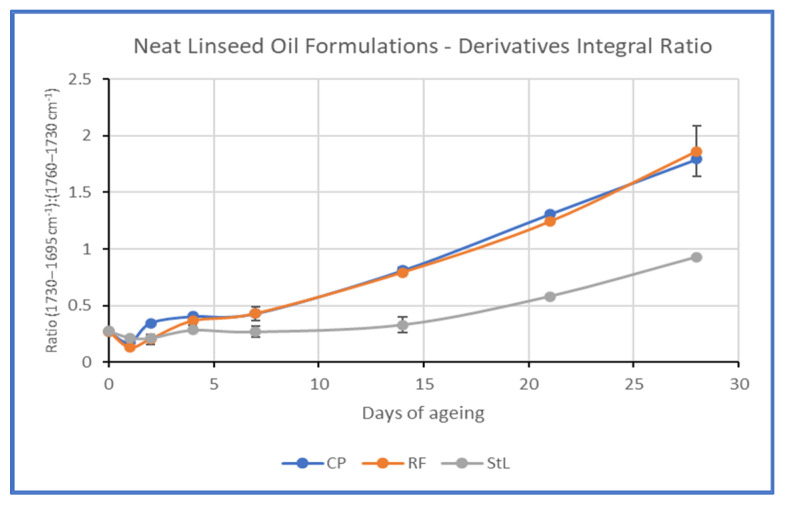
Graphic representation of the integral ratio of the band 1730–1695 cm^−1^ by that of 1760–1730 cm^−1^.

**Figure 11 polymers-15-02567-f011:**
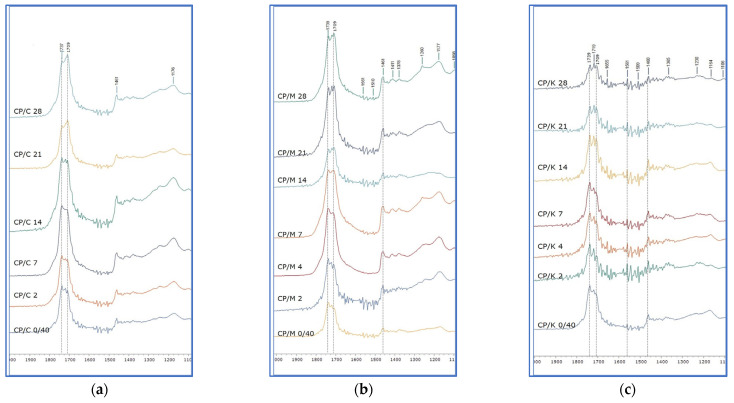
The reflection-FTIR spectra of CP extraction from (**a**) Cotton oil-impregnated mock-ups, (**b**) Montval oil-impregnated mock-ups, and (**c**) Kraft oil- impregnated mock-ups, upon ageing.

**Figure 12 polymers-15-02567-f012:**
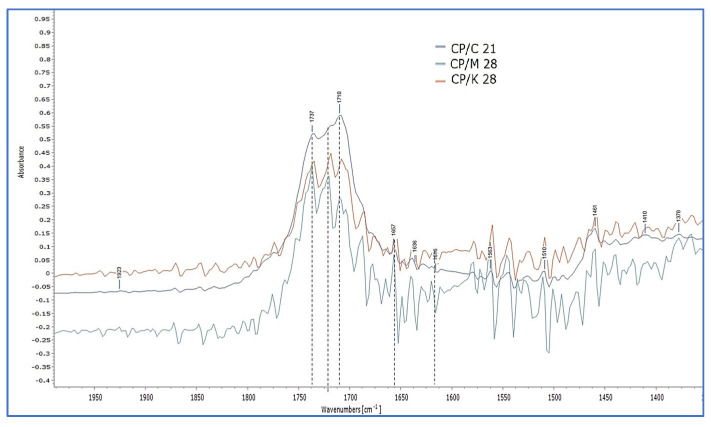
Detail image of the CPs reflection spectra derived from the extraction of C, M, and K impregnated mock-ups, at the final stage of ageing. The formation of several peaks on the band 1700–1550 cm^−1^ is evident.

**Figure 13 polymers-15-02567-f013:**
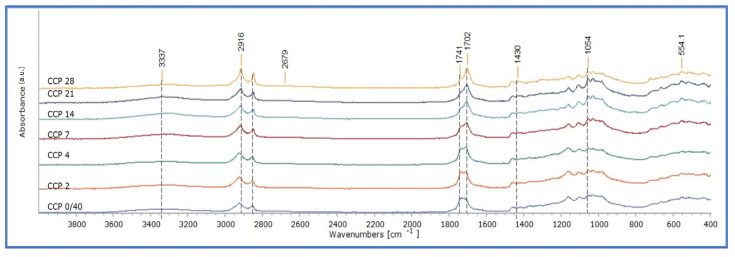
ATR-FTIR spectra of Cotton impregnated mock-ups with cold-pressed linseed oil (CCP).

**Figure 14 polymers-15-02567-f014:**
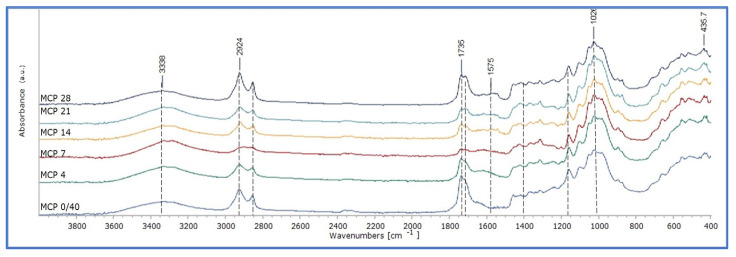
ATR-FTIR spectra of Montval impregnated mock-ups with cold-pressed linseed oil (MCP).

**Figure 15 polymers-15-02567-f015:**
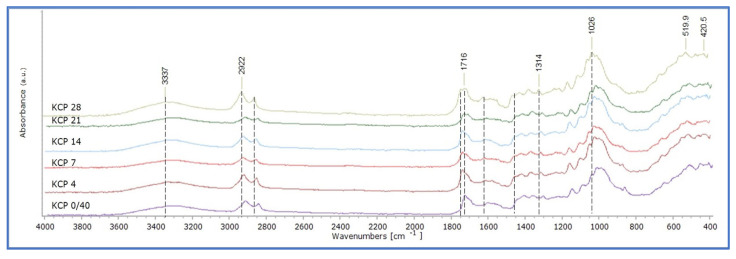
ATR-FTIR spectra of Kraft impregnated mock-ups with cold-pressed linseed oil (KCP).

**Figure 16 polymers-15-02567-f016:**
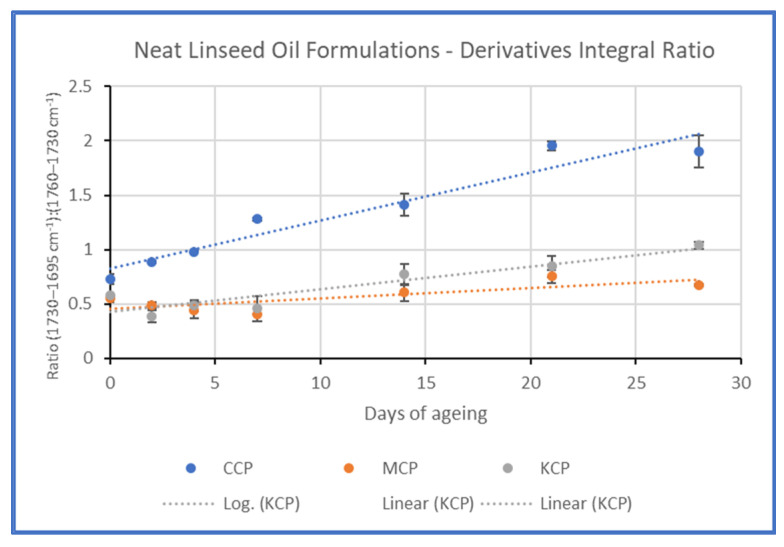
Graphic representation of the calculation of the integral ratio of the band 1730–1695 cm^−1^ by that of 1760–1730 cm^−1^. The trendlines showed the increase in the band 1730–1695 cm^−1^ was higher for Cotton mock-ups, followed by Kraft, and then by Montval at a lower degree.

**Figure 17 polymers-15-02567-f017:**
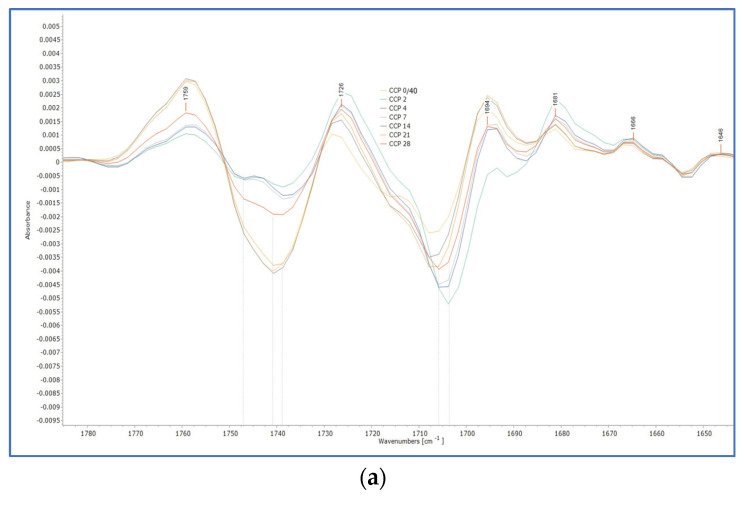
The second derivatives on the band 1750–1700 cm^−1^ for all oil-impregnated mock-ups, at all ageing stages: (**a**) Cotton oil-impregnated mock-ups with CP; (**b**) Montval oil-impregnated mock-ups with CP; and (**c**) Kraft oil-impregnated mock-ups with CP.

**Figure 18 polymers-15-02567-f018:**
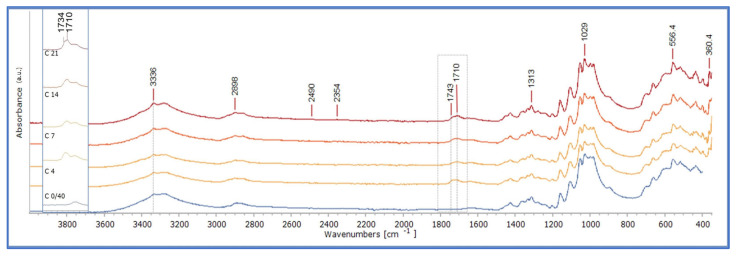
ATR-FTIR spectra of Cotton paper after cold-pressed linseed oil extraction on various stages of artificial ageing (0/40, 4, 7, 14, and 21 days, arrayed from the bottom to the top) and detail.

**Table 1 polymers-15-02567-t001:** The series of mock-ups and their symbols, respectively.

	Symbol	Materials
1	C	Plain Cotton paper
2	M	Plain Montval paper
3	K	Plain Kraft paper
4	CP	Cold-pressed linseed oil
5	RF	Refined linseed oil
6	StL	Stand oil
7	CCP	Cotton paper impregnated with cold-pressed linseed oil
8	CRF	Cotton paper impregnated with refined linseed oil
9	CStL	Cotton paper impregnated with stand oil
10	MCP	Montval paper impregnated with cold-pressed linseed oil
11	MRF	Montval paper impregnated with refined linseed oil
12	MStL	Montval paper impregnated with stand oil
13	KCP	Kraft paper impregnated with cold-pressed linseed oil
14	KRF	Kraft paper impregnated with refined linseed oil
15	KStL	Kraft paper impregnated with stand oil

**Table 2 polymers-15-02567-t002:** Calculation of the integral ratio of two different absorption bands of the three linseed oil formulations after drying for 40 days by those in fresh form.

Absorption Band (cm^−1^)	CP 0–40/CP 0	RF 0–40/RF 0	StL 0–40/StL 0
1820–1570	2.03	1.42	1.57
1450–400	2.95	1.58	3.14

## Data Availability

The data presented in this study are available on request from the corresponding author. The data are not publicly available yet as they derived from an ongoing PhD research project.

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
