# Peer review of "Oil Media on Paper: Investigating the Interaction of Cold-Pressed Linseed Oil with Paper Supports with FTIR Analysis"

_polymers, 2023, doi:10.3390/polym15112567_

Round 1
Reviewer 1 Report
The manuscript contains interesting results on stability of oil treated paper with a sound experimental set up. The text is not ready for publishing, it looks like an uncorrected draft. There are too many figures of IR spectra, most of them are much to small and it is impossible to read them even after zooming on the screen. Legends are missing or misleading (e.g. CP 0-40 is CP 40 and is CP 0 in the following figures, while CP 0 is shown with another meaning in the first graphs). There is no reference to the supplemental document in the text. In the supplemental document the table caption 4S is used twice. In L38 a reference is missing. The last paragraph of the introduction is written like an abstract while the hypothesis of the work is not explained. The term mock-up is not explained. References to research projects and PhD research must be moved to acknowledgements. Some units must be corrected to SI units (e.g. gsm probably means g/m², what is % (w/v)?). SEM/EDX methodology is not described. How can FTIR spectroscopy detect inorganic substances? Table 2 is refereced in the text but it is missing. Fig 2 and 3:Legends are missing, notable changes can be seen but are not described in the text. In chapter 3.2 abbreviations are used that are not explained in Materials and Methods. In L249 the reference to Fig. 10 must probably be corrected to Fig. 7. Figs 10b, 11, 12, 16, 20 and 21 are much too small. Chapter 5 Patents is misleading. Author contributions starts with template text, many points are marked with X.X.
Author Response
Dear Reviewer,
We would like to thank you for your comments and observations. Most of them were to the point and elucidated issues that needed to be resolved. We have tried to respond to them during the revision of the paper. We suggest that the revision of the paper benefited the quality of the outcome.
However, we need to give a few replies to your questions:
- SEM/EDX methodology is not described: SEM-EDX analysis has been initially applied to get an elemental analysis of the paper pulp content at the first stages of research. Reference to that was irrelevant to the subject matter of the paper and it has been removed. However, the FTIR provided more specific data on paper pulp content.
- How can FTIR spectroscopy detect inorganic substances? Fourier-transformed infrared spectroscopy is a method of analysing both organic and inorganic materials and can provide a breakdown of their molecular makeup. There are numerous publications with FTIR results referring on them. Publications on the inorganic content of the pulp content are the following:
- Gorassini, A., Calvini, P. and Baldin, A., 2008. Fourier Transform Infrared Spectroscopy (FTIR) analysis of historic paper: Documents as a preliminary step for chemometrical analysis, multivariate analysis and chemometry. Applied to Environment and Cultural Heritage. [pdf] Available at: <http://w3.uniroma1.it/cma4ch/08/!abstracts/A-Gorassini-Ora(d3cv7zcw9ref).pdf>
- Calvini, P. and Gorassini, A., 2002. FTIR-deconvolution spectra of paper documents. Restaurator, 23(1), pp.48-66.
Reviewer 2 Report
Dear Authors,
your research on the ageing behaviour of linseed oil on paper substrates is interesting. I am really curious to know what can happen on the long time (after 40 days).
The paper, however, needs to be enriched with larger explanation on the occuring oxydation reactions.
The analysis of real samples would be an interesting integration to the theoretical study, together with different ageing conditions/time.
At the moment the paper is a good report of the results, but the discussion section needs to be implemented also with references to the works on ageing oils of other researchers (e.g Izzo Francesca, Miliani Costanza, Fuster-Lopez Laura, etc, etc).
For suggestions on point that could be discussed in depth see also the attached pdf.
Best Regards

Author Response
Dear reviewer,
Thank you for your comments and observations. They were focused and they highlighted parts of the text that need more details and work in order to present the results of our research work adequately. We have tried to respond to most of them. We suggest that the revision was of the benefit of this paper.
However, we have to reply to a few of your comments:
1. Your research on the ageing behaviour of linseed oil on paper substrates is interesting. I am curious to know what can happen in the long time (after 40 days): The mock-ups were air-dried for 40 days in the dark, at 22oC and 57%RH, for practical reasons. It was impossible to handle them in a humid (for oil-impregnated mock-ups) or liquid stage (for neat oils) and submit them to accelerating ageing conditions. Thus, we decided to start the experiment from a “dry” stage. Trials showed that 40 days were enough to get them in a “touch dry” condition (drying of oils is a long-term procedure). Natural ageing for longer was not part of the experimental methodology.
2. The analysis of real samples would be an interesting integration to the theoretical study, together with different ageing conditions/time.
We need to mention that accessing original art works and transferring them aways from the institutions they belong to has a lot of limitations due to insurance and safety reasons. Unfortunately, Covid-19 restriction until recently made access to work even more difficult. Ideally, we needed a collection of works that varied in the time of creation and storage conditions. The works that we had access to, showed small changes in the bands that we investigate, indicating a limited extent of damage. So, we need to find more works to investigate and publish the results in the future. However, the aim of this work was to investigate the input of linseed oil formulations and paper pulp content to the development of chemical reactions, as well as the possible interactions. We hope that the results would provide explanations on the variations recorded on the condition of the oiled areas of the works. This is the novelty of this research work, as there are no similar publications.
3. The conditions of ageing refer to protocols of ageing for papers recommended by the Library of Congress, they have been used in other published research work, and they provided results comparable to the damages recorded to works of art [9, 16, 17, 25]. However, the temperature of 80oC has been also used in research work on linseed oil, where the authors report that provides satisfying results [18]. In the same work, there is a reference that inert conditions do not provide indicative results.
Taking also into consideration the results published by the Library of Congress [25], accelerating ageing in airtight vessels provides adequate ageing results, close to those that occur in real conditions. This methodology is applied in the presence of oxygen.
4. Finally, 24 scans for FTIR analysis provided adequate results for the majority of the mock-ups.
Reviewer 3 Report
The authors describe the results of ATR-FTIR and Reflectance FTIR studies and show the influence of linseed oil on the development of chemical changes in oiled paper areas during the ageing process.
Their results indicate that:
- Linseed oil is the dominant factor in the chemical reactions occurring in the paper during the ageing process.
- It is possible to relate the chemical changes of the paper-linseed oil system to changes occurring in linseed oil preparations.
- The content of cellulose fibres and pulp significantly affects the ageing of the paper-linseed oil system.
Since the authors aim to apply their results to paper conservation processes, the manuscript needs a comparison of spectra from actual historical objects treated with linseed oil to those measured. If the authors have such results at their disposal, the presentation of these results would undoubtedly enhance the quality of the manuscript.
Also, the layout of the manuscript, in which only the spectra are presented, without any tables or figures, is overwhelming. I suggest moving some of the graphs to the SI. And tables such as S1 and S2 should be moved from the SI to the article. Graphics showing the test samples, measurement system, etc., would be an excellent addition to this article.
In summary, the article is interesting and worth publishing, but after the above changes. At that moment, it is presented unattractively and will be lost in the crowd of other similar publications.
Author Response
Dear reviewer,
We would like to thank you for your comments and observations. They were well-spotted and they expressed constructive criticism over the content of the paper. We have tried to respond to them during the revision of the paper, which really benefited the outcome of our work.
However, we have to reply to the following:
1. We need to mention that accessing original artworks and transferring them away from the institutions they belong to has a lot of limitations due to insurance and safety reasons. Unfortunately, Covid-19 restrictions until recently made access to works even more difficult. Ideally, we needed a collection of works that varied in the time of creation and storage conditions. The works that we had access to, showed small changes in the bands that we investigate, indicating a limited extent of damage. So, we need to find more works to investigate and publish the results in the future. However, the aim of this work was to investigate the input of linseed oil formulations and paper pulp content to the development of chemical reactions, as well as the possible interactions. We hope that the results would provide explanations for the variations recorded in the condition of the oiled areas of the works. It is really a problematic issue that conservators have to deal with, due to the diversity of the extent of damage on the oiled areas of the paper support. This is the novelty of this research work, as there are no similar publications. To provide data that could benefit condition assessment of the works as a first step.
Round 2
Reviewer 1 Report
The manuscript has improved compared to the first version. The text is not ready for publishing, it still looks like an uncorrected draft which causes unnecessary work for a reviewer. Many images are much to small and it is impossible to read them even after zooming on the screen. Legends have improved. In the introduction the objectives of this work and the hypothesis are missing, but they are written in the first paragraph of the conclusions. The term mock-up is not explained. References to research projects must be moved to acknowledgements or deleted. Some units must be corrected to SI units, if you can´t avoid usting % (w/v), better express it in SI units such as g/ml. Figs 8, 9, 9S, 10, 12, 17a,b,c are much too small. Author contributions still starts with template text, many points are marked with X.X.
L66 add SI unit for 80°
L77 delete first part of sentence ‘In the framework of a research project’
L108ff please move this text to the introduction
L117 use SI units consequently, such as ml instead of mL
L122 close brackets after ‘days’
L126 correct to ‘have not been submitted to’, is submitted the word you mean or is it subjected?
L129 express % (w/v) in a better way with SI units to be clear
L142 insert space in 1 ml, do this throughout the text with all units
L149 decide for one of these words ‘conducted/performed’
L171ff is text for the Results and Discussion section
L202 the reference to Fig. 1 is probably wrong
L243ff this paragraph needs to be supported by some references
Fig 2ff all absorbance spectra are lacking a scale on the absorbance axis, please add an indication what the scale represents
L290 delete plurals s in ‘14th day’
L291 add ‘after 28 days’ and ‘.’ at the end of this sentence
L352f please explain how cold pressed linseed oil would react with CaCO3 and add a sound reference for this
L485 correct to ‘artificial weathering’
L515f correct to 'The outcome of this work’
L528 delete template text
Author Response
Dear reviewer,
Thank you for your comments and observations. They were focused and pointed out issues to be resolved within the text and graphs. We tried to respond to all of them, hoping that now the content of the paper has been improved.
Only in a few cases, the numbering of the lines and comments did not respond to our version of our file. I wish we had located all of them correctly. Finally, we did not make any changes to the following comments:
The term mock-up is not explained: The term mock-up is commonly used to describe “a model or replica, used for instructional or experimental purposes”. We have come across this term in numerous scientific publications. In the text, we have described the way mock-ups were prepared and what purpose they serve.
Many points in authors’ contributions are marked with X.X.: In our version, we cannot see any XX.
Reviewer 2 Report
Dear Author,
I appreciated the improvements done. Now the text is clear, readable and focuses on some aspects that cannot be discussed in depth in the previous paper.
Author Response
Thank you very much for your comments and your decision to suggest the publication of our work.
Reviewer 3 Report
As for me, the present form of the manuscript is suitable for publication in Polymers.
Author Response

(The authors gave the same response as above.)
